# Philadelphia Chromosome-Positive Leukemia in the Lymphoid Lineage—Similarities and Differences with the Myeloid Lineage and Specific Vulnerabilities

**DOI:** 10.3390/ijms21165776

**Published:** 2020-08-12

**Authors:** Lukasz Komorowski, Klaudyna Fidyt, Elżbieta Patkowska, Malgorzata Firczuk

**Affiliations:** 1Department of Immunology, Medical University of Warsaw, Nielubowicza 5 St, 02-097 Warsaw, Poland; lkomorowski@wum.edu.pl (L.K.); klaudyna.fidyt@wum.edu.pl (K.F.); 2Postgraduate School of Molecular Medicine, Medical University of Warsaw, Trojdena 2a St, 02-091 Warsaw, Poland; 3Department of Hematology, Institute of Hematology and Transfusion Medicine, Indiry Gandhi 14, 02-776 Warsaw, Poland; epatkowska@ihit.waw.pl

**Keywords:** leukemia, CML, B-ALL, TKIs, drug targets, p190, p210, treatment, B cell, BCR-ABL1

## Abstract

Philadelphia chromosome (Ph) results from a translocation between the breakpoint cluster region (*BCR*) gene on chromosome 9 and ABL proto-oncogene 1 (*ABL1*) gene on chromosome 22. The fusion gene, *BCR-ABL1*, is a constitutively active tyrosine kinase which promotes development of leukemia. Depending on the breakpoint site within the *BCR* gene, different isoforms of BCR-ABL1 exist, with p210 and p190 being the most prevalent. P210 isoform is the hallmark of chronic myeloid leukemia (CML), while p190 isoform is expressed in majority of Ph-positive B cell acute lymphoblastic leukemia (Ph+ B-ALL) cases. The crucial component of treatment protocols of CML and Ph+ B-ALL patients are tyrosine kinase inhibitors (TKIs), drugs which target both BCR-ABL1 isoforms. While TKIs therapy is successful in great majority of CML patients, Ph+ B-ALL often relapses as a drug-resistant disease. Recently, the high-throughput genomic and proteomic analyses revealed significant differences between CML and Ph+ B-ALL. In this review we summarize recent discoveries related to differential signaling pathways mediated by different BCR-ABL1 isoforms, lineage-specific genetic lesions, and metabolic reprogramming. In particular, we emphasize the features distinguishing Ph+ B-ALL from CML and focus on potential therapeutic approaches exploiting those characteristics, which could improve the treatment of Ph+ B-ALL.

## 1. Introduction

BCR-ABL1 is a constitutively active tyrosine kinase encoded by the fusion gene consisting of the breakpoint cluster region (*BCR*) and the ABL proto-oncogene 1 (*ABL1*). The reciprocal translocation between the two chromosomes, t(9;22) (q34;q11), results in the formation of the abnormally short chromosome 22, commonly known as the Philadelphia chromosome (Ph) [1]. It was first discovered and described by Nowell and Hungerford as a recurring characteristic small chromosome in neoplastic cells in patients suffering from chronic myeloid leukemia (CML) [2]. Depending on the breakpoint site, three isoforms varying in length can be distinguished: the shorter variant of 185–190 kDa molecular weight (p190), the longer variant of 210 kDa molecular weight (p210), and the longest variant p230, which is rare and not discussed in this review [3]. A diagram showing the structural differences between p190 and p210 is presented in Figure 1. The exact mechanism of Ph chromosome generation and the determinants of the formation of specific variants are not fully elucidated. Nevertheless, non-homologous end joining, single-strand annealing [4] and recombinase-activating gene (*RAG*) activity [5] have been proposed as mediators of these processes.

Ph chromosome occurs mainly in CML and B cell acute lymphoblastic leukemia (B-ALL). The diseases are different in terms of molecular and clinical features. CML is derived from hematopoietic stem cells and the expression of BCR-ABL1 in leukemic stem cells is sufficient to initiate and promote leukemogenesis. In the majority of cases, CML is diagnosed at a chronic, mild phase, and is easily manageable with tyrosine kinase inhibitors (TKIs), the very first targeted drugs that were developed in the late 1990s. Introduction of TKIs to CML treatment protocols was a major breakthrough in the management of the disease and the fate of patients. In rare cases, CML is diagnosed in more advanced, accelerated, or blast phase of myeloid or lymphoid origin. The transition to the blast crisis may also occur during the treatment with TKIs due to developing resistance [8,9]. Ph+ B-ALL is one of the subtypes of acute lymphoblastic leukemia (ALL) and its frequency increases with patients’ age. It originates from B cell committed progenitors. Apart from BCR-ABL1 translocation, there are numerous other genetic alterations already present at diagnosis. It is usually diagnosed in more advanced stage as compared to CML and TKIs are less effective due to primary or secondary resistance, resulting in low overall survival of Ph+ B-ALL patients [8,10].

The incidence of different BCR-ABL1 isoforms vary between the diseases—p210 is found in approximately 95% of CML cases while p190 is present in 70% of Ph+ B-ALL patients [11]. Although both isoforms contain intact ABL1-derived kinase domain (KD), recent findings report significant differences in the interactomes, signaling, and the subcellular localization of the two isoforms [6,12]. Moreover, research of the last decade uncovers a plethora of linage-specific mutations and signaling pathways, which confer unique vulnerabilities and may serve as novel therapeutic targets [13,14,15,16,17].

Here we describe the main clinical and genetic features of CML and Ph+ B-ALL and present current treatment protocols of the diseases, with the main focus on the role of TKIs. We highlight major differences in signaling mediated by the two main BCR-ABL1 isoforms. We also provide an overview of recently published research on lineage-specific differences in crucial signaling pathways other than BCR-ABL1. Finally, in the light of recent findings, we discuss novel, targeted, lymphoid lineage-specific treatment options, both already tested in clinical trials and those that have shown effectiveness in preclinical models.

## 2. Genetic Alterations and Characteristics of Different Types of Leukemia with BCR-ABL1 Translocation

Even though the BCR-ABL1 is the main onco-driver in CML and Ph+ B-ALL, they are very distinct diseases, with different sets of accompanying mutations. In this paragraph we compare characteristic features and genetic profiles of both diseases and their changes during progression.

### 2.1. Chronic Myeloid Leukemia

CML is a myeloproliferative disease of hematopoietic cells in which the major event needed for malignant transformation is BCR-ABL1 translocation [18]. It constitutes 15% of all diagnosed adult leukemia cases in the US (1.95 new cases annually per 100,000 people) [19]. In Europe it affects one in 100,000 people each year, with the average age at diagnosis 64 years [20]. CML is usually diagnosed in a manageable, chronic phase (CP), but it can progress into accelerated phase (AP) or a fatal blast phase (BP). Main molecular alterations in CML include hyperactivation of signaling pathways that lead to uncontrolled cell proliferation and decreased apoptosis [21,22].

#### 2.1.1. Chronic Phase CML

CP is the first and mild phase of CML. During CP malignant cells are able to undergo close to normal maturation and regulation, showing almost no symptoms in patients until the leukemic cells outgrow healthy cells significantly [23]. CML is believed to initiate from one or more hematopoietic stem cells (HSCs) that have acquired leukemic phenotype, called leukemia stem cells (LSCs). These cells are often resistant to therapy and responsible for renewal of the leukemic cells population after therapy discontinuation. Various pathways are responsible for better survival of LSCs including enhanced DNA repair mechanisms. Recently, novel methods of selective targeting of LSCs by inhibition of the DNA repair mechanisms are being uncovered [24].

Researchers have been trying to determine the exact surface antigens characteristic of LSCs, enabling to distinguish LSCs from the normal stem cells, with varying results [25]. A recently published single cell molecular analysis of the bone marrow (BM) cells isolated from CML patients in CP led to the discovery of a distinct, BCR-ABL1 positive Lin^−^CD34^+^CD38^−/low^CD45RA^−^cKIT^−^CD26^+^ LSC population. Those cells were therapy-resistant, molecularly quiescent and primitive [26]. Another single cell analysis of CML LSCs throughout the course of therapy uncovered that LSCs which persisted after prolonged TKIs treatment presented transforming growth factor β (TGF-β)- and tumor necrosis factor α (TNF-α)-related gene expression signature [27]. This property was not only limited to Ph+ LSCs, as Ph-negative (Ph-) HSCs in CML patients poorly responding to therapy also showed increased levels of TGF-β and TNF-α signaling, indicating disruption of the whole microenvironment [27].

CML in CP has been considered a disease driven by a single *BCR-ABL1* genetic aberration with a small number of additional mutations, present only in 10% of patients [28]. Recently, two independent single cell analyses of newly diagnosed CML CP patients showed that mutations in methylation-related genes are present in 25–30% of patients. Identified mutations were found in genes such as additional sex combs like 1 (*ASXL1*), Tet methylcytosine dioxygenase 2/3 (*TET2/3*), (K)-specific demethylase 1A (*KDM1A*), mutS homolog 6 (*MSH6*), DNA methyltransferase 3A (*DNMT3A*) [29,30], although their significance has not been investigated so far.

#### 2.1.2. Advanced Phases of CML

Even though CML in CP is now a manageable disease, in up to 5–10% of cases it progresses to more aggressive AP and finally to a fatal BP, either of myeloid (MBP) or lymphoid (LBP) lineage (70% and 30% of cases, respectively [31]). This process is accompanied by the accumulation of additional genetic aberrations. Malignant cells in BP lose the ability to differentiate, leading to their rapid clonal expansion and accumulation of immature myeloid or lymphoid progenitors [22].

Despite efforts to elucidate precise causes and mechanisms responsible for CML progression, the answer is still elusive. One of the reasons is the genetic complexity and heterogeneity of BP with numerous coexisting mutations and signaling disruptions, making it hard to point out specific events that advance the disease toward BP. The most apparent feature is elevation of BCR-ABL1 expression during progression into BP, causing increased genetic instability and accumulation of additional mutations [32]. Approximately 70–80% of BP patients have additional chromosomal mutations as compared to only 10–30% of patients with additional mutations found in CP [28,29,30]. Apart from common BCR-ABL1 point mutations leading to TKIs resistance, MBP and LBP have distinct mutational landscape. The most common alterations in MBP are those of epigenetic regulator *ASXL1* and tumor suppressor genes, such as tumor protein 53 (*TP53;* up to 20% of cases) and runt-related transcription factor 1 (*RUNX1*; ≈ 40% of cases). In LBP, the most frequent mutations occur in cyclin dependent kinase inhibitor 2A/B gene (*CDKN2A/B*; 50% of cases) and IKAROS family zinc finger 1 (*IKZF1*; 55% of cases) genes [32,33,34]. *CDKN2A/B* gene encodes proteins involved in p53 and cyclin dependent kinases (CDKs) regulation, indicating its role as tumor suppressor [35]. Product of *IKZF1* gene is a critical regulator of B cell differentiation and maturation, crucial for B lineage commitment of precursor cells [36]. A recent study linked *IKZF1* and *CDKN2A/B* mutations in LBP to increased RAG activity and RAG-mediated events [37], suggesting RAG expression as a factor contributing to the transition to CML LBP. Below we briefly summarize recent findings related to mechanisms and major regulators of CML progression.

A serine-threonine phosphatase 2A (PP2A), responsible for dephosphorylation of many key cell regulators, has been identified as one of the factors in CML advancement to BP. PP2A has been shown to act as a tumor suppressor in some malignancies and its inactivation has been observed in CML MBP cell line (K562) and primary cells [38,39]. In most cases, PP2A downregulation is not caused by mutations in PP2A-encoding gene, as they are rarely observed in leukemias [38]. Conversely, the mechanism turned out to be BCR-ABL1-mediated upregulation of the SET nuclear proto-oncogene (SET) protein, a physiological inhibitor of PP2A [39], thus proving unrestrained BCR-ABL1 activity in triggering BP.

MYC proto-oncogene (MYC), a transcription factor belonging to the basic-helix-loop-helix-leucine zipper family, has also been linked to BCR-ABL1 activity and CML progression [40,41]. A study utilizing a mouse model with inducible expression of BCR-ABL1 in hematopoietic stem and progenitor cells, mirroring CML progression in humans, revealed elevated MYC expression during progression of CML to BP. This observation has also been made for human MBP primary material [41]. In vitro inhibition of the bromodomain and extra terminal (BET) proteins, which control MYC levels and functioning, significantly reduced colony formation of primary cells and cell lines, in both MBP and LBP, revealing MYC role in BP of both lineages [41]. Interestingly, analysis of *BCR* promoter controlling BCR-ABL1 expression revealed that binding of MYC upregulates BCR-ABL1 expression, creating an upregulation feedback loop [40]. MYC silencing in CML BP cell lines caused significant BCR-ABL1 downregulation and cell death, proving MYC a critical factor in CML progression and maintenance [40].

Another driver of CML progression seems to be calcium-calmodulin-dependent kinase IIγ (CaMKIIγ). It has been identified as a regulator of signaling pathways in leukemia cells, but its role in BP initiation was not clear. Gu et al. provided proof that CaMKIIγ is critical in CML course as its aberrant activation was observed in MBP but not CP. CaMKIIγ deletion resulted in cell cycle arrest in LSCs, but not in normalHSCs. Accordingly, CaMKIIγ overexpression in CML MBP cell line K562 resulted in significant increase in the proliferation rate of the malignant cells. Immunoprecipitation of CaMKIIγ revealed that it phosphorylates p27Kip1 protein, a CDK inhibitor, resulting in a decrease of nuclear p27Kip1 content, and acceleration of cell cycle [42].

In recent years, epigenetics emerged as another factor in CML progression. Several studies presented BCR-ABL1-triggered disruption in methylation patterns during progression to BP [43,44]. Exact mechanisms of methylome alterations are unknown, but a possible explanation could be repression of histone 3 lysine 4 demethylase, retinoblastoma binding protein (RBP2), by BCR-ABL1 [45]. A recent comprehensive study involving whole-genome analyses and chromatin immunoprecipitation has uncovered a potential role of polycomb repressive complex 1 and 2 (PRC1/2) proteins in both MBP and LBP. In BC PRC1 triggered repression of tumor suppressors that regulate cell death pathways, cell cycle, and inflammation, such as early growth response protein 1 (EGR1) or nuclear receptor subfamily 4 group A member 2 (NR4A2). Furthermore, the component of PRC2, histone-lysine N-methyltransferase, enhancer of zeste homolog 2 (EZH2), directed a repressive DNA hypermethylation (histone 3 lysine 27 tri-methylation) in BC cells, leading to silencing of genes responsible for myeloid differentiation [46]. It is worth mentioning that direct comparison of MBP and LBP methylation patterns showed remarkable similarities in their transcriptomes and methylomes [46], suggesting methylation-independent mechanisms in lineage determination of CML cells entering BP. An interesting observation was higher upregulation of several pathways related to metabolism in MBP compared to LBP, including mammalian target of rapamycin (mTOR) complex 1 (mTORC1), which is an energy sensor and a master growth regulator [46].

### 2.2. Philadelphia Positive B Cell Acute Lymphoblastic Leukemia

ALL affects 1.64 and 1.28 people per 100,000 yearly in the USA [19] and Europe [47], respectively. It is a heterogeneous disease of either T or B cell lineage (25% and 75% of all adult ALL cases, respectively) with many distinct subtypes, characterized by the accumulation of immature lymphoid progenitors with various genetic alterations. Unlike in CML, ALL blasts comprise heterogeneous subclones at various stages of B cell differentiation with equal capacity of leukemia propagation, therefore ALL lacks stem cell hierarchy [48]. Ph+ B-ALL is one of the high-risk B-ALL subtypes and occurs in approximately 30% of adult ALL cases [49]. Ph+ B-ALL is also diagnosed in children, but at much lower rate, constituting 2–4% of childhood ALL [49]. In Ph+ B-ALL, the p190 isoform of BCR-ABL1 is the most frequent [50]. In contrast to CML, BCR-ABL1 translocation alone is insufficient for malignant transformation, with various complex additional mutations necessary for Ph+ B-ALL development [50].

Similar to CML LBP, the most recurrent mutations in Ph+ B-ALL occur in *IKZF1* gene. Over 70% of Ph+ B-ALL patients harbor *IKZF1* loss-of-function, usually resulting from mono-allelic deletions (bi-allelic deletions occur in 13–15% of cases [51,52]) or from the expression of dominant-negative isoforms, with the IK6 isoform being the most prevalent [53]. IK6 expression has been shown to be correlated with increase in BCR-ABL1 activity during disease progression and worse prognosis to patients [54]. Similar to CML LBP, RAG expression has been shown to be elevated in Ph+ B-ALL and responsible for exons 3–6 deletion, resulting in the generation of the IK6 isoform [53,55].

Deletions in *CDKN2A/B* gene is another frequent mutation present in approximately 45% of Ph+ B-ALL cases [56]. Frequency of *CDKN2A/B* deletions in Ph+ B-ALL has been shown to be independent of patient age [57] and its higher occurrence has been observed in patients after a relapse [35].

Deletions in paired box 5 (*PAX5*) gene, encoding one of the B cell-specific transcription factors regulating B cell differentiation, are present in approximately 30–40% of Ph+ B-ALL cases [56,58]. Importance of *PAX5* deletions was underlined by an in vivo study, showing that expression of p190 alongside monoallelic *PAX5* deletion in B cell precursors caused B-ALL in 90% of animals, with numerous genetic alterations in the remaining *PAX5* gene allele [59]. Another interesting observation was made through genetic analysis of all B-ALL subtypes revealing interesting link of *PAX5* mutations with occurrence of *CDKN2A* alterations. As 100% of analyzed adults harboring *PAX5* deletions harbored *CDKN2A* deletions as well [60], it indicates the importance and convergence of these two genes in B-cell leukemias.

Several less frequent mutations in Ph+ B-ALL can also be distinguished. A genetic study of 97 Ph+ B-ALL patients revealed recurring alterations in the following genes: B-cell translocation gene 1 (*BTG1*; 18%), retinoblastoma protein 1 (*RB1*; 14%), early B-cell factor 1 (EBF1; 13%) and translocation-Ets-leukemia virus (ETV6; 5%) [56]. *EBF1* and *ETV6* are involved in B cell differentiation, signal transduction, and lineage commitment [61,62]. Together with frequent mutations of *PAX5* and *IKZF1*, it suggests that deletions of B cell-specific transcription factors are main events required for Ph+ B-ALL development [63,64]. The most prevalent mutations identified in CML in CP, MBP, LBP, and Ph+ B-ALL are summarized in Table 1.

## 3. Similarities and Differences in Signaling Mediated by p190 and p210 BCR-ABL Variants

BCR-ABL1 owes its activity to the ABL1 KD, constitutively activated by its fusion with BCR. ABL1, together with ABL2, make up the ABL non-receptor tyrosine kinases family. Both kinases have some overlapping functions in regulation of cell growth and proliferation, but also have some unique roles [65]. Knockout of ABL1 in mice caused neonatal lethality in approximately 50% of mice, while mice deprived of ABL2 were viable, indicating a major role of ABL1 in animal survival [65]. Activity of both ABL kinases is regulated by changes in their structure, with two main states—active “open” state and inactive “closed” state [65]. ABL1 kinase contains three nuclear localization motifs and one nuclear export signal, regulating its subcellular localization [65]. These domains are crucial for ABL1 activity, as its subcellular localization defines effects of its activation. In the cytoplasm, it increases proliferation rate and survival of cells, while in the nucleus it leads to cell cycle arrest and cell death promotion [66]. At least one normal ABL1 allele is usually present in cells expressing BCR-ABL1, acting as a tumor suppressor, as its knockout in CML CP cells leads to increased survival of cells in vitro and development of much more aggressive leukemia in mice [66]. Even though ABL1 can have opposing effects based on its localization, the BCR-ABL1 oncoprotein is normally positioned in the cytoplasm despite the presence of the nuclear localization signals [67]. In the cytoplasm BCR-ABL1 associates with membranes or cytoskeleton, allowing it to exert its growth-promoting, pro-leukemic effects [67].

The BCR-ABL1 tyrosine kinase activates several major signaling pathways in cells. Two of such are the RAS/mitogen-activated protein kinases (RAS/MAPK) pathway and nuclear factor kappa-light-chain-enhancer of activated B cells (NF-κB), which activity promotes abnormal cell proliferation, cell cycle progression, and survival [68]. Additionally, RAS/MAPK pathway together with BCR-ABL1 activate phosphoinositide 3-kinases/protein kinase B (PI3K/AKT) and mammalian target of rapamycin (mTOR), inactivating apoptotic machinery and prolonging cell survival [68]. Another signaling cascade prominently activated by BCR-ABL1 is the Janus kinase-signal transducer and activator of transcription proteins (JAK/STAT) pathway. The main consequence of its activation is induction of cytokine independence and improved survival [68].

Even though BCR-ABL1 isoforms, p190 and p210, have identical kinase domains [6], their preferential distribution in distinct diseases suggests signaling differences. Development of an accurate mouse model allowed for better insight into oncogenic potential of different BCR-ABL1 isoforms [18]. In an in vivo study, mouse BM was transduced with constructs expressing p190 or p210 and injected into tail vein of irradiated, recipient mice. In one group donor BM was primed with 5-fluorouracil prior to harvest, which has been shown to stimulate HSCs and its niche [69]. Injection of primed BM caused development of CML-like disease in both p190 and p210 injected mice, but introduction of non-primed cells caused significantly different diseases. Non-primed p190 bearing cells caused an aggressive B lymphoid leukemia in most recipient mice, while p210 induced more mild leukemia with CML-like characteristics. In some cases, those CML-like diseases later progressed into acute B lymphoid leukemia, reconstituting the natural course of CML and its progression into BP [18]. These results revealed apparent differences in leukemogenic potential between BCR-ABL1 isoforms, however it did not explain the causes behind it.

In 2017, two comprehensive, quantitative proteomic studies comparing p190 and p210 signaling were published, shedding more light on differences in p190 and p210 signaling [12,70]. To ensure the same molecular background for their studies, both groups utilized an in vitro approach using the IL3-dependent murine hematopoietic Ba/F3 cell line expressing either p190 or p210 and compared their BCR-ABL1 interactomes and phosphoproteomes. Both studies revealed sets of common BCR-ABL1 interacting proteins consisting mostly of known BCR-ABL1 interactors, such as CRK proto-oncogene (CRK), Cbl proto-oncogene (CBL), PI3K regulatory subunit 2 (PIK3R2), son of sevenless homolog 1/2 (SOS1/2), SH2 domain-containing inositol 5′-phosphatase 1/2 (SHIP1/2), and growth factor receptor bound protein 2 (GRB2).

Importantly, two sets of proteins with preferential or exclusive interaction with a specific isoform have been discovered. One of the most significant differences observed in signaling mediated by p190 and p210 isoforms was the differential regulation of STAT proteins. While STAT1 was activated mainly by p190, STAT3 and STAT5 preferentially bound with p210 isoform. P210 phosphorylated STAT5 at five different sites, including the major one responsible for dimerization and transcriptional activation. Although STAT5 phosphorylation also occurs in p190-bearing leukemia, no direct interaction between STAT5 and p190 was observed, suggesting that it is mediated by another upstream kinase—JAK2 [12,70]. This is in agreement with previous reports of JAK2 indispensability in leukemogenesis driven by p190 but not by p210 isoform [71].

Another important distinction between p190- and p210-mediated signaling is differential binding and activation of proto-oncogene tyrosine-protein (SRC) kinases. SRC family kinase LYN and its interacting partner—hematopoietic cell specific LYN substrate 1 (HCLS1)—showed stronger interaction with p190 compared to p210, and LYN phosphorylation was stronger in p190-expressing cells [70]. Accordingly, LYN activity was previously shown to be necessary for CML progression into LBP but not MBP [70]. On the other hand, p210 more potently activated other SRC family kinases, FYN and LCK, as well as MAPK1/2 and tyrosine kinase FES. The latter kinase was previously shown to drive myeloid differentiation in CML [72].

Another converging observation of the two proteomic studies was the different subcellular localization of the p190 and p210 variants. While p190 bound preferentially to the cytoskeleton remodeling proteins (Wiskott–Aldrich syndrome family members, docking protein 1 (DOK1), NCK adaptor protein (NCK1/2), GRB2, SHIP2), p210 interacted with proteins located in membrane proximity: CBL, Cbl proto-oncogene B (CBLB), SHIP1/2, SHC adaptor protein 1 (SHC1), and ubiquitin associated and SH3 domain containing B (UBASH3b). The latter is an ubiquitin-binding membrane adaptor protein with a phosphatase activity, so it could serve as a negative regulator. Further structural studies revealed that interaction of p210 with UBASH3b is dictated by the subcellular localization. The removal of membrane-binding domains from p210 decreased both membrane association and interaction with UBASH3b [6]. Recent studies confirmed a negative regulatory role of UBASH3b in p210 expressing cells [73]. UBASH3b knockdown caused hyperphosphorylation of several proteins crucial for leukomegenesis, such as p210 and STAT5, in sites important for their activity [74]. Lack of UBASH3b also increased p210 interaction with STAT5 and CRK, showing that UBASH3b partially downregulates p210 interactome [74].

## 4. Tyrosine Kinase Inhibitors as a Core Treatment of Different Types of Ph+ Leukemia

The introduction of TKIs has revolutionized the treatment of Ph+ leukemia. These targeted drugs, in comparison with conventional chemotherapy, present lower toxicity and better direct action against specific oncogenic pathways [75]. Imatinib mesylate (also known as STI571) was the first TKI developed by rational drug design in the late 1990s, and approved for medical use by FDA (Food and Drug Administration) in 2001. It is an ATP-competitive kinase inhibitor and its mechanism of action relies on blocking of the ATP binding to the inactive BCR-ABL1 conformation and preventing tyrosine phosphorylation. This process inhibits leukemia cells’ proliferation and leads to their apoptosis. Imatinib, which has been assigned the first-generation TKI, is effective in many patients, yet still approximately 40% of patients with CML-CP quit receiving imatinib due to failure and/or intolerance [76,77]. Therefore second- and third-generation TKIs have been developed to provide greater efficacy in patients who were resistant or intolerant to imatinib. TKIs approved for the treatment of CML and Ph+ B-ALL, their major targets, and clinical indications are presented in Table 2.

### 4.1. TKIs for the Treatment of CML

#### 4.1.1. Treatment of CML in Chronic Phase

Four TKIs are approved by the FDA and the European Medicines Agency (EMA) for the use as monotherapy in the 1st line treatment of CML patients: imatinib, nilotinib, dasatinib, and bosutinib. The fifth TKI, radotinib, is so far registered only in South Korea. Additionally, ponatinib has been approved for CML patients resistant to two or more TKIs and these with the BCR-ABL1 T315I mutation. Molecular response to the treatment is assessed by quantitative measurements of BCR-ABL1 transcript levels at 3, 6, and 12 months after therapy initiation, and is provided in percentage, according to the international scale [9]. Response criteria for the 1st line treatment are defined by the European LeukamiaNet (ELN) 2020 Recommendations [9]. Reduction of BCR-ABL1 level below 0.1%, achieved after 12 months of TKI treatment, is defined as the major molecular response (MMR, or MR) and predicts CML-specific survival close to 100%. The choice of a particular TKI depends on the presence of mutation(s) in the *BCR-ABL1* gene, comorbidities, contraindications, and costs. A suboptimal response to two or more TKIs should result in consideration of allogeneic hematopoietic stem cell transplantation (allo-HSCT) [9].

The International Randomized Study of Interferon and STI571 (IRIS) was the first one that reported clinical benefit of imatinib in the 1st line treatment of CML, presenting 5-year progression-free survival (PFS) and overall survival (OS) of 80–90% and 90–95%, respectively [9,88]. Since the approval of imatinib, life expectancy of the majority of CML patients does not deviate from the general population. Imatinib is a safe drug without life-threatening adverse effects [88,89]. Additionally, it is cost-effective, as generic imatinib showed similar efficacy and safety [90,91]. The efficacy of second-generation TKIs in clinical trials was compared to imatinib arm. Although the deep molecular response (DMR, BCR-ABL1 level below 0.01%) could be achieved faster after treatment with the second-generation TKIs, it did not translate to better effectiveness (PFS and OS) [92,93,94]. The multicenter study recently published by Efficace et al. demonstrated better health-related quality of life in patients treated with dasatinib than imatinib in frontline therapy [95]. The treatment response and survival outcomes of CML patients to the 1st line TKIs are summarized in [96]. A recent noteworthy fact is that approximately 40–50% CML patients with deep and durable molecular responses can safely discontinue therapy [91,97,98].

#### 4.1.2. Treatment of CML in Advanced Phases

The onset of CML in advanced phases is rare. According to the European Treatment and Outcome Study (EUTOS), at the diagnosis stage of CML the incidence of AP is 3.5% and BP is 2.2% [99]. Additionally, TKIs failure in CP and progression to advanced phases is uncommon. In the IRIS trial, the estimated 10-year cumulative incidence of BP was 7.9% [88]. Despite low incidence, advanced phases of CML still remain a challenge in therapy and have poor prognosis [100]. As BCR-ABL1 transcript level is the propulsion for disease progression, the most effective management of BP is its prevention. Therefore, rapid and deep BCR-ABL1 reduction is the most important therapeutic goal [100].

Therapy of BP consists of intensive chemotherapy in combination with a selected TKI and prompt qualification for allo-HSCT, or non-intensive approaches in elderly patients. Intensive chemotherapy in MBP is based on anthracycline and cytosine arabinoside. However, other drugs combinations including etoposide, carboplatin, fludarabine, 5-azacitidine, or decitabine are also used [101]. Unfortunately, a return to CP phase was reported only in approximately 10% of MBP patients. More treatment options are available for LBP and higher response rates are observed. Combinations of vincristine and prednisone or hyperfractionated cyclophosphamide, vincristine, adriamycin, dexamethasone (HCVAD), as well as cytosine arabinoside, methotrexate, 6-thioguanine, and 6-mercapropurine allowed to achieve 15–50% responses in LBP. Survival and response rates are better after treatment with TKIs than with conventional chemotherapy, but the duration of response remains short with a median survival of 3–10 months [100]. The results of selected clinical studies conducted in patients with advanced phases of CML are presented in Table 3.

### 4.2. TKIs in the Treatment of Ph+ ALL

Over the last decade, there has been great progress in understanding of the pathogenesis as well as in treatment options in Ph+ B-ALL. Introduction of TKIs to the therapy improved the long-term outcomes by allowing more patients to undergo allo-HSCT and by achieving durable relapse-free survival without undergoing allo-HSCT in less fit patients. Such interventions allowed to improve the outcomes of Ph+ B-ALL patients that in adults are now comparable to those observed in Ph- B-ALL. Still the 5-year OS usually does not exceed 50%, hence novel treatment options are urgently needed [115]. In children, the survival rates of Ph+ B-ALL patients are still poorer as compared to other subtypes [116]. Currently, three different TKIs are approved by FDA and by EMA for the treatment of Ph+ B-ALL in adults: imatinib, dasatinib, and ponatinib. In Ph+ B-ALL patients, TKIs are administered together with chemotherapy regimens (see below). Table 4 summarizes clinical trials of TKI-including regimens in adults with Ph+ B-ALL.

For imatinib, as well as for the second- and third-generation TKIs, the results of small, single-arm studies are only available with no direct comparison between different TKIs in adults. In pediatric Ph+ B-ALL, the recently published results of randomized clinical trial revealed better efficacy of dasatnib over imatinib, with 4-year OS of 88.4% and 69.2%, respectively [129]. An industry-sponsored study comparing imatinib at a dose of 600 mg with ponatinib at a dose of 30 mg is ongoing (NCT03589326). Based on the results from single-arm studies and the propensity score matching, second- and third-generation TKIs were superior to imatinib considering treatment outcomes and clinical benefits for Ph+ B-ALL [127,130]. In contrast, a non-randomized, retrospective comparison of imatnib vs. second-generation TKIs reveals no difference [131].

The first line treatment of Ph+ B-ALL usually consists of a TKI in combination with chemotherapy in all treatment stages involving induction, consolidation, maintenance, and central nervous system (CNS) prophylaxis. Induction therapy usually includes vincristine, prednisone, anthracycline with or without L-asparaginase and cyclophosphamide. Consolidation includes 6–8 courses containing high-dose methotrexate, cytosine arabinoside, L-asparaginase, cyclophosphamide [120,132]. In relapsed, treatment-refractory Ph+ B-ALL patients whose blast cells express antigens available for immunotherapy, monoclonal antibodies (mAbs) against CD19 (blinatumomab) and CD22 (inotuzumab ozogamicin) are used with relatively good efficacy, similar to Ph- B-ALL [133,134]. 

Allo-HSCT with a strong graft vs. host leukemia (GvHL) effect is a very important form of immunotherapy in the treatment of Ph+ B-ALL patients [135]. However, high incidence of treatment-related mortality (up to 40%) and high rates of acute and chronic graft vs. host disease are limiting the widespread use of allo-HSCT [136]. Therefore, in the era of TKIs combination therapy, the importance of allo-HSCT in Ph+ B-ALL patients is debatable, especially in patients who quickly achieve deep molecular responses [137]. Patients treated with intensive chemotherapy in combination with ponatinib do not require allo-HSCT, due to high percentage of complete molecular response (CMR, BCR-ABL1 undetectable). The recently reported 3-year OS of those patients reached 79% [127,138].

### 4.3. Resistance to TKIs in CML and Ph+ B-ALL

Different generations of TKIs have significantly improved the outcomes for Ph+ patients around the globe, however intrinsic and acquired resistance to TKIs is still an issue that is being addressed by researchers. The landscape of TKIs resistance can be divided to those related to mutational (ABL1 KD mutations) and non-mutational (BCR-ABL1 independent) changes in leukemic cells. Considering that TKIs were primarily used for the treatment of CML, many reports describing resistance mechanisms were based on data acquired from CML patients [139,140]. The most common mutational changes are point mutations in a T315 of *ABL1*, known as a gatekeeper residue, which effectively limits the TKIs binding. A variety of point mutations in *ABL1* have been already described, including T315I, M244V, M351T, Y253F/H, F317L, or E255K/V, of which T315I determines resistance to first- and second-generation TKIs [139,141,142]. Major progress in understanding and management of TKIs resistance has been made over the years, while novel approaches are currently focused on better predictions of resistance phenotype to occur. This is possible due to better accessibility to high-throughput approaches such as next generation sequencing (NGS). Within the multicenter NEXT-in-CML study, Soverini et al. presented that NGS is superior to Sanger sequencing in detecting low BCR-ABL1 mutation variant allele frequency (3–20%), which may account for early relapse and development of other mutations [143]. Moreover, patients who neither achieve optimal responses, nor fail to TKIs treatment, were considered as the “warning zone” patients according to the ELN recommendations [144]. Some of those patients had low variant allele frequency mutations, including T315I, and subsequently developed fully resistant phenotype. Hence, low-level mutations in such patients should encourage clinicians for earlier TKIs switch.

While the *ABL1* point mutations-induced resistance can be managed by switching to other TKIs, such approach may cause the selective pressure on leukemic cells and emergence of compound mutants leading to multi-TKIs resistance. T315I concomitant mutations, such as D276G or F359V confer ponatinib resistance [145], despite its proven efficacy towards T315I single mutants [146]. The compound resistance is more likely to occur in high-risk patients with BP-CML and Ph+ B-ALL rather than CP-CML [147], and more studies are needed to identify better treatment options for such patients.

Despite sufficient inhibition of BCR-ABL1 protein, many patients can still experience failure of the treatment due to development of BCR-ABL1-independent mechanisms of TKIs resistance. Thus far, these non-mutational mechanisms have been more extensively studied in CML than in Ph+ ALL, and are described in great detail elsewhere [139,148,149]. Alternative activation of the BCR-ABL1 downstream pathways, including JAK/STAT, SRC, MAPK, or PI3K seems to be crucial in mediating this particular resistance phenotype, which explains better efficacy of less selective TKIs [150,151,152]. In regard to Ph+ B-ALL, similar mechanisms are likely to account for the BCR-ABL1 independent resistance. Indeed, both RAF/MAPK [153] and AKT/mTOR [154] pathways were reported to mediate imatinib-induced non-mutational resistance in Ph+ B-ALL cell lines and primary cells. In addition to BCR-ABL1 downstream targets, the leukemia microenvironment plays significant role in development of TKIs resistance. Accordingly, Mallampati et al. [155] showed the role of mesenchymal stem cells (MSCs) in mediating TKIs resistance in Ph+ ALL. Imatinib-treated MSC increased the production of pro-survival molecules, including IL-7, which in turn activated JAK1 and JAK3 and provided survival advantage for B-ALL cells.

### 4.4. Novel Tyrosine Kinase Inhibitors Tested in Clinical Trials

Considering persistent incidences of TKIs resistance and relapses in Ph+ leukemias, researchers have focused on designing novel inhibitors that could overcome treatment failures and reduce common adverse effects. Such interventions led to the discovery of asciminib, an allosteric inhibitor of BCR-ABL1. In contrast to other TKIs it does not bind to ATP site, but to myristoyl site of ABL1, thus providing efficacy towards all known single ABL1 mutants [156]. The first in human phase I clinical trial evaluating the effects of asciminib in Ph+ leukemias is still ongoing (NCT02081378), however primary results from heavily pretreated CML patients showed very promising efficacy [157]. Despite encouraging preclinical and clinical data, several issues remain to be addressed in asciminib monotherapy: (1) emergence of resistance due to point mutations in myristoyl-binding site, (2) ineffectiveness towards compound mutant cells. Eide at al [158] described several mutations in the myristoyl-binding site accounting for asciminib-resistant phenotype, which was completely abrogated by addition of nilotinib or ponatinib. More importantly, the combination of asciminib and ponatinib effectively diminished the viability of compound mutant cells and prolonged the survival of leukemia-bearing mice, thus showing great promise for resistant and relapsed Ph+ patients [158]. Accordingly, many clinical trials are currently recruiting CML and Ph+ ALL patients to evaluate the efficacy of asciminib in combination with other TKIs (NCT03595917, NCT03578367, NCT02081378).

## 5. Unique Features and Lineage-Specific Drug Sensitivities of Ph+ Lymphoid Leukemia

Apart from the preferential occurrence of p210 in CML and p190 in Ph+ B-ALL, and the consequent differences in signaling, there are other factors responsible for distinct characteristics of the two diseases. Many of these differences are associated with lymphoid or myeloid lineage commitment. Noteworthy, CML in LBP, which is mainly driven by p210 isoform, molecularly resembles more Ph+ B-ALL than CML. In this paragraph we present recent findings reporting distinct traits of B-lymphoid Ph+ leukemia, determining its unique drug vulnerabilities that could be utilized in therapy.

B cell precursors commit to lymphoid lineage at a very early stage of differentiation, which is followed by several steps leading to development of cells expressing the pre-B cell receptor (pre-BCR) complex. At this stage cells undergo two step selection: (1) positive—cells that express proper pre-BCR heavy chain are preserved; (2) negative—autoreactive cells are removed due to aberrantly strong activation of pre-BCR downstream signaling with self antigens [159]. Signals defining survival and proliferation of cells are transduced through immunoreceptor tyrosine-based activation motifs (ITAM) present in the cytoplasmic tails of Igα and Igβ, which are components of the pre-BCR complex. Either insufficient or excessive activation of pre-BCR signaling leads to pre-B cells apoptosis. Even though Ph+ B-ALL cells lack both Igα and Igβ on their surface and should undergo apoptosis during positive selection, BCR-ABL1 activates pre-BCR downstream signaling and prevents this process to occur. Importantly, imbalanced, excessive signaling by BCR-ABL1 can also lead to Ph+ B-ALL cells’ death via negative selection [160]. This indicates existence and importance of mechanisms keeping activation of those pathways on moderate level. One of such mechanisms is overexpression and oncogenic role of phosphatase and tensin homolog deleted on chromosome 10 (PTEN), a PI3K-AKT signaling inhibitor, in Ph+ B-ALL cells [16]. This observation is somewhat surprising, as PTEN is considered a tumor suppressor and its mutations are found in various cancers, however, they are absent in B-ALL patients [16]. Most malignant cells require upregulated PI3K-AKT activity and BCR-ABL1 assures its constant activation, yet in B-ALL cells it must be precisely controlled in order to avoid negative selection. Genetic or pharmacologic inhibition of PTEN in Ph+ B-ALL cells caused hyperactivation of AKT and prominent cell death [16], uncovering the unique role of PTEN in Ph+ B-ALL survival. Another way Ph+ B-ALL cells avoid negative selection is the expression of surface inhibitory receptors such as platelet endothelial cell adhesion molecule 1 (PECAM1), CD300A, and leukocyte-associated immunoglobulin-like receptor 1 (LAIR1), which recruit the inhibitory phosphatase SHIP1, calibrating the strength of BCR-ABL1 signaling. Expression level of these receptors correlates with poor outcome of B-ALL patients. Ablation of either of these proteins led to hyperactivation of SRC and extracellular signal-regulated (ERK) kinases, resulting in senescence, cell cycle arrest, and decreased clonogenic potential of Ph+ B-ALL cells, but not in CML cells [160]. One of the approaches that already showed efficacy toward Ph+ B-ALL primary cells in vitro was the inhibition of SHIP1 phosphatase using 3 α aminocholestane, causing hyperactivation of pre-BCR signaling [160]. In summary, the above reports confirm that Ph+ B-ALL cells have functional mechanisms of negative selection that can be utilized in therapy.

Another unique feature of B cells is a constant state of energy depletion and modulation of energy-utilization pathways [13,14]. It has been postulated that the low energetic reserves serve as a mechanism preventing hyperactivation and excessive proliferation of autoreactive and activated B cells [15]. One of such mechanisms is negative regulation of glucose uptake and metabolism [13,14,15]. Recently published results revealed a crucial role of two B lineage-specific transcription factors, IKZF1 and PAX5, in the regulation of B cell metabolism, explaining frequency of their inactivating mutations in Ph+ B-ALL [13]. Reconstitution of wild-type variants of IKZF1 or PAX5 in primary B-ALL cells caused downregulation of several glucose transporters (GLUT1, GLUT3, and GLUT6) and glucose utilization effectors (hexokinase 2/3, HK2/3; glucose-6-phosphate dehydrogenase, G6PD), paralleled by the upregulation of direct and indirect glucose transport inhibitors (nuclear receptor subfamily 3 group C member 1, NRC3C1; thioredoxin-interacting protein, TXNIP; cannabinoid receptor type 2, CNR2), leading to energy crisis and cell death [13]. Energy stress sensor protein, 5′AMP-activated protein kinase (AMPK) has been identified as mediator of IKZF1 and PAX5 effect on cell metabolism, as its pharmacological inhibition in Ph+ B-ALL cells caused similar effects as reconstitution of those transcription factors, causing energy depletion and cell death [13]. Another interesting observation was that primary B-ALL cells had over 10-fold-lower levels of these glucose-related proteins than CML cells [13]. Low expression of the proteins involved in glucose utilization in Ph+ B-ALL is most likely due to residual activity of both PAX5 and IKZF1 in Ph+ B-ALL, as the vast majority of their inactivating mutations in Ph+ B-ALL are monoallelic [53,59]. Accordingly, even partial downregulation of PAX5 in the BM resulting from monoallelic deletion promoted p190-driven leukemia with much higher penetrance and earlier disease onset than in mice with normal PAX5 expression (90% vs. 13% incidence) [59].

Cancer cells usually maintain activation of the alternative glucose utilization pathways, such as pentose phosphate pathway (PPP), which provides NADPH and salvages oxidative stress [161]. Xiao et al. showed that normal and malignant B cells repress PPP through PAX5 and IKZF1, but maintain PPP at minimal level, providing additional evidence of the key role of PAX5 and IKZF1 in shaping B cell metabolism [17]. PP2A was identified as the key enzyme responsible for the PPP preservation in Ph+ B-ALL cells, indicating its essential role for survival of these cells. PP2A represses glycolytic activity and redirects glucose metabolism towards PPP, providing NADPH to fuel antioxidant machineries [17]. Expression levels of PP2A are higher in primary Ph+ B-ALL cells compared to CML cells [17]. Knockdown of PP2A or its pharmacological inhibition lowered NADPH levels, reduced expression of antioxidant genes (e.g., catalase, *CAT*; superoxide dismutase 2, *SOD2*; peroxiredoxin 2/4, *PRDX2/4*), increased reactive oxygen species (ROS), and resulted in apoptosis of Ph+ B ALL cells. None of these effects was observed in CML cells, confirming that these processes are B-lineage specific [17].

Some unique vulnerabilities of Ph+ B-ALL result from the B cell-specific effects of BCR-ABL1 signaling. A recent study reported that BCR-ABL1-mediated upregulation of the atypical protein kinase C (aPKC), normally involved in cell polarity regulation, contributes to Ph+ B-ALL transformation and maintenance [162]. Elevated expression of aPKCι (human homolog of murine aPKCλ) was detected in human primary Ph+ B-ALL cells. Murine B cell progenitors with aPCKλ deletion upregulated B cell-specific transcription factors PAX5 and IKZF1 through epigenetic mechanisms and induced B cell differentiation. Accordingly, the effects of aPKCλ deletion, decreased proliferative potential and increased apoptosis, were more pronounced in Ph+ B-ALL than in CML cells. Furthermore, in vivo studies confirmed that aPKCλ is needed for BCR-ABL1-induced leukemogenesis [162]. All these results show that suppression of B cell differentiation is crucial for the oncogenic transformation of B cell progenitors with BCR-ABL1 and uncovers mediators of the suppression as putative drug targets.

Another recent study revealed γ-catenin, a member of the catenin family involved in the regulation of cell adhesion, as a B cell-specific target in Ph+ leukemia [163]. Mice injected with the Ph+ BM cells derived from γ-catenin deficient donors either did not develop disease (41% of recipients) or developed CML-like disease (36%), while almost all recipients of Ph+ control BM cells developed B-ALL (90%) [163]. γ-catenin was also necessary for Ph+ B-ALL cells maintenance, as inducible γ-catenin deletion in mice with detectable Ph+ B-ALL cells in blood and BM led to symptom-free survival throughout the study (60 days), while control animals succumbed to the disease within 40 days post injection of leukemic cells [163]. Analysis of gene expression after γ-catenin deletion in Ph+ BM cells showed significant downregulation of WNT signaling target genes, including MYC and survivin. MYC expression was reduced to 50% and prevented B-ALL development, suggesting that the expression of both MYC alleles is necessary for leukemic transformation [163]. Furthermore, mechanistic studies showed that in Ph+ B cell progenitors, BCR-ABL1 and LYN kinases phosphorylated γ-catenin at different sites, and only the phosphorylated protein induced MYC expression and promoted leukemia growth. Β-catenin, another member of catenin family which has been previously found to be crucial for CML initiation and progression [164], was dispensable for Ph+ B-ALL development. In summary, catenins are another example of differential, lineage-specific regulation of Ph+ leukemia development.

## 6. Concluding Remarks and Future Directions

Even though CML and Ph+ B-ALL are both driven by the constitutively active BCR-ABL1 tyrosine kinase, their course and prognosis differ significantly. While in the great majority of cases CML can be kept under control with the use of TKIs, these targeted drugs used in monotherapy are less effective in the management of Ph+ B-ALL. Combinations of TKIs with other chemotherapy result in 5-year OS not exceeding 50%. Therefore, better understanding of signaling pathways activated specifically in Ph+ B-ALL may indicate new treatment options and improve patient outcomes. Furthermore, it is of particular importance for elderly patients who are not eligible to intensive chemotherapy. Novel B lineage-specific vulnerabilities and putative therapeutic targets identified over the last decade are summarized in Figure 2A. One of the major molecular differences between CML and Ph+ B-ALL is the preferential occurrence of the longer, p210 BCR-ABL1 variant in CML, and the shorter, p190 variant in Ph+ B-ALL. Recent, in-depth characterization of the signaling pathways triggered by p190 and p210 gave insight into differential signaling of the two BCR-ABL1 isoforms, suggesting isoform-specific subcellular localization as a main determinant of their interactomes and downstream signaling [12,70]. Importantly, these studies presented specific oncogenes that could be targeted specifically in Ph+ B-ALL, including JAK2 as well as SRC kinase LYN [12,70]. Other studies have shown that combined phosphorylation of γ-catenin by BCR-ABL1 and LYN kinase, both of which are essential in Ph+ B-ALL, promotes expression and transactivation of MYC oncogene, another key player of BCR-ABL1-driven leukemia initiation and propagation in B cell progenitors [163]. Targeting of SRC, JAK2, as well as other B cell-specific targets are presented in Figure 2A in a left panel. Another B cell-specific strategy that has proved effective in preclinical models is triggering the negative selection-associated apoptosis. Importantly, it could be achieved by inhibition of phosphatases, such as PTEN [16] or SHIP-1 [16], which are responsible for balanced downstream kinase signaling (Figure 2A, middle panel). Metabolic differences between Ph+ myeloid and lymphoid malignant cells revealed another B cell-specific vulnerability. Activation of direct and indirect glucose transport inhibitors (TXNIP, CNR2) as well as blocking of AMPK or PP2A caused rapid energy depletion and oxidative stress-mediated cell death specifically in Ph+ B-ALL cells (Figure 2A, right panel) [13,17]. Finally, initiation of B cell differentiation program via selective depletion of aPKC proved effective in triggering cell cycle arrest and mediating apoptosis in Ph+ B-ALL. Importantly, many of the aforementioned targets are druggable with small molecule inhibitors which already proved their efficacy in monotherapy or in combinations with TKIs.

Apart from intracellular targets associated with distinct metabolism and signaling, Ph+ B-ALL can also be targeted by its unique, B cell-specific surface antigens. Immunotherapies with mAbs specific to CD19, CD20, CD22, CD38, as well as chimeric antigen receptor (CAR)-modified T cell therapy, have recently revolutionized the treatment of hematological malignancies. Attempts have been made to combine some of these immunotherapies with TKIs for the treatment of relapsed/refractory (R/R) Ph^+^ ALL or LBP CML. For instance, anti-CD22 antibody-drug conjugate, inotuzumab ozogamicin, is tested in combination with bosutinib in a phase I/II clinical trial (NCT02311998), and so far demonstrates significant efficacy and good tolerance to the treatment [165]. Daratumumab, anti-CD38 mAb approved for the treatment of multiple myeloma, showed anti-leukemic effects in combination with ponatinib in R/R Ph+ ALL in a single case study [166] and is further investigated in a clinical trial in pediatric and young adult R/R B-ALL (NCT03384654). Blinatumomab, a bi-specific T cell engager targeting concomitantly CD3 and CD19 molecules, proved its anti-leukemic effects in R/R Ph^+^ ALL as monotherapy [133] and in combination with second- and third-generation TKIs [167]. An ongoing clinical trial is also testing dasatinib and blinatumomab combination as a frontline therapy for adult Ph^+^ B-ALL (NCT02744768). Examples of ongoing clinical trials testing novel treatments in Ph+ B-ALL are presented in Figure 2B.

In summary, we have recently witnessed great progress in understanding the differences in signaling pathways triggered by the oncogenic BCR-ABL1 kinase in different types of leukemia. In addition, other BCR-ABL1-independent vulnerabilities of Ph+ B-ALL were discovered and identified as potential drug targets. In many cases, these newly identified pathways can be blocked using clinically available inhibitors. Further preclinical and clinical studies combining these new drugs with TKIs and/or modern immunotherapies may contribute to a breakthrough in Ph+ B-ALL treatment and bring real benefits to patients.

## Figures and Tables

**Figure 1 ijms-21-05776-f001:**
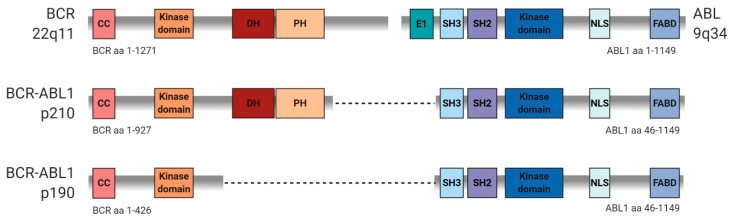
Structural differences between p210 and p190 isoforms of the BCR-ABL1 fusion gene, based on [6] and [7]. Abbreviations: CC, coiled-coil; DH, Dbl-homology; E1, ABL1 exon 1; FABD, F-actin binding domain; NLS, nuclear localization signal; PH, Pleckstrin-homology; SH2/3, SRC homology 2/3.

**Figure 2 ijms-21-05776-f002:**
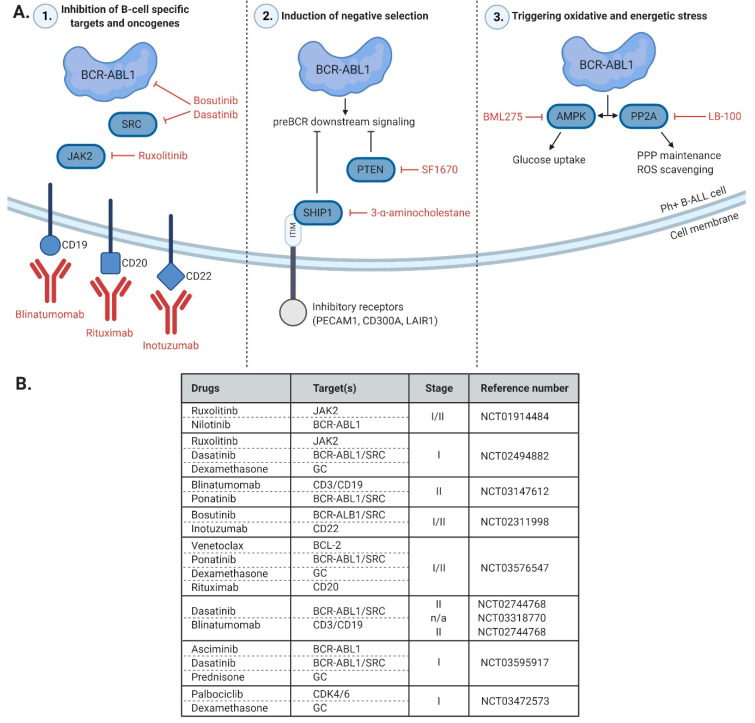
Novel therapeutic targets in Ph+ B-ALL. The figure description is in Section 6. (**A**) Ph+ B-ALL specific therapeutic targets (blue color) and their inhibitors (red color). Abbreviations: AMPK, 5′AMP-activated protein kinase; BCR-ABL1, breakpoint cluster region-ABL proto-oncogene 1; BML275, small molecule AMPK inhibitor; ITIM, immunoreceptor tyrosine-based inhibition motif; JAK2, Janus kinase 2; LB-100, small molecule PP2A inhibitor; PP2A, serine-threonine phosphatase 2A; PPP, pentose phosphate pathway; preBCR, preB-cell receptor; PTEN, phosphatase and tensin homolog deleted on chromosome 10; ROS, reactive oxygen species; SF1670, small molecule PTEN inhibitor; SHIP1, SH2 domain-containing inositol 5′-phosphatase 1; SRC, proto-oncogene tyrosine-protein kinase Src. (**B**) Selected ongoing clinical trials testing novel therapies of Ph+ B-ALL. Abbreviations: BCL-2, B-cell lymphoma 2; BCR-ABL1, breakpoint cluster region-ABL proto-oncogene 1; CDK4/6, cycline dependent kinase 4/6; GC, glucocorticoid receptor; JAK2, Janus kinase 2; SRC, proto-oncogene tyrosine-protein kinase Src.

**Table 1 ijms-21-05776-t001:** Most frequent mutations in chronic myeloid leukemia (CML) and Ph-positive B cell acute lymphoblastic leukemia (Ph+ B-ALL) other than BCR-ABL1. Abbreviations: *ASXL1*, additional sex combs like 1; *BCoR*, BCL6 interacting corepressor; *BCoRL1*, BCL6 corepressor like 1; *BTG1*, B-cell translocation gene 1; *CDKN2A/B*, cyclin dependent kinase inhibitor 2A/B; *DNMT3A*, DNA methyltransferase 3A; *EBF1*, early B-cell factor 1; *ETV6*, translocation-Ets-leukemia virus; *IKZF1*, IKAROS family zinc finger 1; *KDM1A*, (K)-specific demethylase 1A; *MSH6*, mutS homolog 6; *PAX5*, paired box 5; *RB1*, retinoblastoma protein 1; *RUNX1*, runt-related transcription factor 1; *TET2/3*, Tet methylcytosine dioxygenase 2/3; *TP53*, tumor protein P53.

Disease	Gene Name	Frequency of Mutations	Gene Function	References
CML CP	*ASXL1*	10%	Epigenetic regulator; a member of the Polycomb group of proteins	[29,30,33,34]
	*TET2/3*	8%	Epigenetic regulator; catalyzes the conversion of methylcytosine to 5-hydroxymethylcytosine	[29,30]
	*DNMT3A*	8%	Epigenetic regulator; methylates CpG sites	
	*KDM1A*	3%	Epigenetic regulator; demethylates H3K4me2	
	*MSH6*	3%	Component of DNA mismatch repair mechanism	
CML MBP	*ASXL1*	40%	Epigenetic regulator; a member of the Polycomb group of proteins	[33,34]
	*RUNX1*	20–40%	Transcription factor; regulates HSCs differentiation	[32,33,34]
	*TP53*	20%	Regulation of cell cycle, DNA repair, apoptosis	[32]
	*BCoRL1*	10%	Apoptosis regulator; interacts with histone deacetylases	[33,34]
CML LBP	*IKZF1*	55%	Transcriptional regulator; regulates B cell development	[32,33,34]
	*CDKN2A/B*	50%	Regulation of cell cycle, apoptosis; inhibits cyclin dependent kinases, stabilizes p53	[32]
	*RUNX1*	25–35%	Transcription factor; regulates HSCs differentiation	[33,34]
	*BCoR*	15–25%	Apoptosis regulator; interacts with histone deacetylases	
Ph+ B-ALL	*IKZF1*	70%	Transcriptional regulator; regulates B cell development	[51,52]
	*CDKN2A/B*	45%	Regulation of cell cycle, apoptosis; inhibits cyclin dependent kinases, stabilizes p53	[56]
	*PAX5*	30–40%	Transcription factor; regulates B cell development	[56,58]
	*BTG1*	18%	Negative regulation of cell proliferation	[56]
	*RB1*	14%	Regulation of cell cycle progression	
	*EBF1*	13%	Transcription factor; regulates B cell development	
	*ETV6*	5%	Transcription factor; regulates development of hematopoietic cells	

**Table 2 ijms-21-05776-t002:** Characteristics of tyrosine kinase inhibitors. Abbreviations: ABL1, ABL proto-oncogene 1; AKT, protein kinase B; BCR, breakpoint cluster region; CaMK2G, calcium/calmodulin-dependent protein kinase type II gamma chain; CDK2, cyclin-dependent kinase 2; c-KIT, proto-oncogene c-KIT; CML, chronic myeloid leukemia; EPH, erythropoietin-producing human hepatocellular receptor; FAK, focal adhesion kinase; FGFR1, fibroblast growth factor receptor; FLT3, fms like tyrosine kinase 3; KIT, proto-oncogene c-KIT; LCK, lymphocyte-specific protein tyrosine kinase; MEK, mitogen-activated protein kinase kinase; PDGFR, platelet-derived growth factor receptor; Ph+ B-ALL, Philadelphia positive B cell acute lymphoblastic leukemia; RET, RET proto-oncogene; SRC, proto-oncogene tyrosine-protein kinase Src; TEC, tyrosine-protein kinase Tec; TKI, tyrosine kinase inhibitor; VEGFR, vascular endothelial growth factor.

TKI Generation	Name	Major Targets	Indications	Mechanism of Action Unique Properties	Side Effects
First generation	Imatinib mesylate [78,79]	BCR-ABL1, PDGFR, c-KIT, EPH	Newly diagnosed adult and pediatric patients with CML in chronic phase, blast crisis, adult patients with newly diagnosed, relapsed or refractory Ph+ B-ALL	ATP-competitive TKI; binds to the inactive conformation of ABL1	Gastrointestinal symptoms, joints pain, skin rash, fatigue (frequent); cardiovascular events (5%)
Second generation	Dasatinib [80,81,82]	BCR-ABL1, PDGFR, c-KIT, EPH, FGFR1, APKK, CDK2, AKT, p38, FAK, SRC, LCK, c-KIT	Imatinib-resistant or intolerant CML and Ph+ B-ALL, newly diagnosed chronic phase CML	ATP-competitive TKI; higher inhibitory potential against BCR-ABL1 than imatinib; binds to the active conformation of ABL1; penetrates to the central nervous system	Pleural effusions (37%); pulmonary arterial hypertension (rare)
	Nilotinib [83]	BCR-ABL1, PDGFR, c-KIT, EPH	Imatinib-resistant or intolerant chronic and accelerated phase CML, newly diagnosed CML	ATP-competitive TKI; better topographical fit for the ABL1 than imatinib; binds to the inactive conformation of ABL1	Cardiovascular events (20%); pancreatitis (5%)
	Bosutinib [84,85]	BCR-ABL1, SRC, LCK, TEC, CaMK2G, PDGFR, c-KIT	Imatinib- or dasatinib- or nilotinib-resistant CML patients	ATP-competitive TKI; binds to the active and inactive conformation of ABL1	Transient diarrhea, nausea, gastrointestinal symptoms (30%)
	Radotinib [86]	BCR-ABL1, PDGFR, c-KIT, SRC	Approved in South Korea for CML chronic phase in patients newly diagnosed or with insufficient response to other TKIs	ATP-competitive TKI; structurally similar to imatinib and to nilotinib	Fatigue, nausea, asthenia (rare)
Third generation	Ponatinib [87]	BCR-ABL1 including BCR-ABL1^T315I^, RET, FLT3, KIT, FGFR, VEGFR1, VEGFR2, PDGFR, SRC, EPH, Auora kinases	CML and Ph+ B-ALL patients with the BCR/ABL1^T315I^ mutation or resistant to two or more TKIs	ATP-competitive TKI; binds to the inactive conformation of ABL1	Gastrointestinal symptoms, joints pain, skin rash, fatigue (frequent); cardiovascular events (30%)

**Table 3 ijms-21-05776-t003:** Selected trials in advanced phase of CML, based on [102]. Abbreviations: AP, accelerated phase; BP, blast phase; CCyR, complete cytogenic response; CHR, complete hematologic response; CML, chronic myeloid leukemia; FLAG-Ida, fludarabine, arabinoside cytosine, G-CSF, idarubicin regimen; HR, hematologic response; LBP, lymphoid blast phase; MBP, myeloid blast phase; MCyR, major cytogenetic response; MMR, major molecular response; OS, overall survival; PFS, progression-free survival. Definitions of Hematologic, Cytogenetic, and Molecular Response are provided in [103].

Drug(s)	Number of Patients	Hematologic Response	Cytogenetic/Molecular Response	Survival
First generation TKI in AP at CML diagnosis				
Imatinib [104]	87	CHR 85%	CCyR 47% MMR 34%	6-years PFS 48%
First generation TKI in BP at CML diagnosis				
Imatinib [105]	92	MBP: CHR 24% LBP: CHR 35%	MCyR 12% CCyR 10%	Median survival 7 months
Second-generation TKI in AP at CML diagnosis				
Nilotinib [106] Dasatinib [106]	66	CHR 97%	CCyR 84% MMR70%	7-years OS 87%
Second- and third-generation TKI in AP CML after imatinib and/or other TKI				
Dasatinib [107]	317	CHR 47–52%	CCyR 32–33%	2-years OS 63–72%
Bosutinib [108]	79	HR 57%	MCyR 40%	4-years OS 59%
Ponatinib [109]	83	HR 55%	CCyR 24%	1-year OS 84%
Second- and third-generation TKI in BP CML after imatinib and/or other TKI				
Dasatinib [110]	149 MBP 61 LBP	HR 28% MBP HR 38% LBP	MCyR 27% MBP MCyR 46% LBP	2-years OS 24–28%MPB 16–21% LBP
Ponatinib [109]	52 MBP 10 LBP	HR 29% MBP HR 40% LBP	MCyR 19% MBP MCyR 40% LBP	-
Nilotinib [111]	105 MBP 31 LBP	HR 60% MBP HR 59% LBP	MCyR 38% MBP MCyR 52% LBP	2-years OS 32%MPB 10% LBP
Third-generation TKI in BP CML after dasatinib or nilotinib failure or intolerance				
Ponatinib [109]	38 MBP 24 LBP	HR 32% MBP HR 29% LBP	MCyR 18% MBP MCyR 29% LBP	2-years OS 29% MPB 29% LBP
TKIs in combination with chemotherapy				
HCVAD+imatinib/dasatinib [112]	42 LBP	CHR 90%	CCyR 58%	Median survival 17 months
Different TKIs+chemotherapy [113]	195 LBP/MBP	HR 64%	CCyR 29%	Median survival 12 months
FLAG-Ida+ponatinib [114]	17 BP	CHR 17%	MCyR 52%	1-year OS 45.8%

**Table 4 ijms-21-05776-t004:** Selected clinical trials in adults and young adults with Ph+ B-ALL (based on [117,118]). Abbreviations: allo-HSCT, allogeneic hematopoietic stem cell transplantation; allo-HSCT rate, the rate of patients receiving allo-HSCT; CMR, complete molecular remission; OS, overall survival.

Regimen	N	Age Median (Range)	CMR Rate	Allo-HSCT Rate	OS Rate
Imatinib					
Imatinib + intensive chemotherapy [119]	169	42 (16–64)	-	72%	38% (4 years)
Imatinib + intensive chemotherapy [120]	133	45 (21–9)	23% (two cycle)	65%	46% (5 years)
Imatinib + non-intensive chemotherapy [120]	135	49 (18–59)	29% (two cycles)	62%	46% (5 years)
Dasatinib					
Dasatinib + corticosteroids [121]	53	54 (24–77)	15% (day 85)	42%	31% (20 months)
Dasatinib + intensive chemotherapy [122]	72	55 (21–0)	65% (overall)	17%	46% (5 years)
Dasatinib + non-intensive chemotherapy [123]	71	69 (55–83)	24% (consolidation)	10%	36% (5 years)
Nilotinib					
Nilotinib + intensive chemotherapy [124]	90	47 (17–71)	77% (3 months)	63%	72% (2 years)
Nilotinib + chemotherapy [125]	47	66 (55–85)	30% (induction)	-	47% (5 years)
Bosutinib					
Bosutinib monotherapy [126]	24	59 (24–84)	-	-	Median OS 3.6 months
Ponatinib					
Ponatinib + intensive chemotherapy [127]	37	51 (27–75)	78% (overall)	24%	80% (2 years)
Ponatinib + corticosteroids [128]	44	> 60 years	45% (8 weeks)	-	-

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
