# Peer review of "Philadelphia Chromosome-Positive Leukemia in the Lymphoid Lineage—Similarities and Differences with the Myeloid Lineage and Specific Vulnerabilities"

_ijms, 2020, doi:10.3390/ijms21165776_

Round 1

Reviewer 1 Report

There are numerous translocations between the BCR and the ABL oncogene in the so called Philadelphia chromosome.  This review discusses the differences between the P210 isoform of ABL which is the hallmark of chronic myeloid 19 leukemia (CML), while p190 isoform is expressed in majority of Ph-positive B cell acute 20 lymphoblastic leukemia (Ph+ B-ALL) cases. Tyrosine kinase inhibitors are used with success in CML patients but the Ph+ B-ALL patients often become drug resistant. 

With this in mind the authors review recent discoveries related to differential signaling pathways mediated by different BCR-ABL1 isoforms, lineage-specific genetic lesions and metabolic reprogramming. They focus on potential therapeutic approaches exploiting those characteristics, which could improve the treatment of Ph+ B-ALL.  Thus this review is of benefit to clinicians and scientists.

This review is quite comprehensive and thorough with numerous tables and one figure. 

It is fine as is but would benefit from one more figure showing the various break points and which domains of the ABL protein are encompassed in the p210 vs p190 proteins.  Amplification of the region between 190 and 210 would be quite informative.

Author Response

There are numerous translocations between the BCR and the ABL oncogene in the so called Philadelphia chromosome.  This review discusses the differences between the P210 isoform of ABL which is the hallmark of chronic myeloid 19 leukemia (CML), while p190 isoform is expressed in majority of Ph-positive B cell acute 20 lymphoblastic leukemia (Ph+ B-ALL) cases. Tyrosine kinase inhibitors are used with success in CML patients but the Ph+ B-ALL patients often become drug resistant.

With this in mind the authors review recent discoveries related to differential signaling pathways mediated by different BCR-ABL1 isoforms, lineage-specific genetic lesions and metabolic reprogramming. They focus on potential therapeutic approaches exploiting those characteristics, which could improve the treatment of Ph+ B-ALL.  Thus this review is of benefit to clinicians and scientists.

This review is quite comprehensive and thorough with numerous tables and one figure.

It is fine as is but would benefit from one more figure showing the various break points and which domains of the ABL protein are encompassed in the p210 vs p190 proteins.  Amplification of the region between 190 and 210 would be quite informative.

Authors: We agree with the suggestion that an additional figure presenting structural differences between p190 and p210 would be beneficial for better understanding of the article. We prepared the figure and included it in the revised version of the manuscript (Figure 1, L 46-49).

Reviewer 2 Report

This is a well-written well-referenced review of some molecular features of CML and Ph+ ALL along with TKI-directed therapies for these diseases. I congratulate authors for preparing such an organized comprehensive review. 

I have a few comments:

1) Suggest considering moving section 4 (starting at line 413) to an earlier part of the manuscript. This section may fit better prior to treatment strategies

2) Line 309-310: Mostly, addition of TKIs to chemotherapy regimens improves long-term outcome (eg, relapse-free survival and OS) in Ph+ ALL and not necessarily/significantly the rate of first complete remission. Majority of patients with Ph+ ALL achieve a complete morphologic remission without TKIs, but they have high relapse rate. 

3) Lines 334, 335, 336: Addition of rituximab to chemotherapy has been mainly studied in Ph- B-cell ALL (Ph+ ALL excluded from studies). Although theoretically rituximab therapy appears effective regardless of Ph status, studies have excluded these patients. 

4) Lines 350, 351: Total body irradiation is typically done for allogeneic stem cell transplantation and not for auto-HSCT

5) Lines 596-597: The statement "these targeted drugs used in monotherapy are ineffective in the management of Ph+ B-ALL". TKI monotherapy can be effective in this disease, but the efficacy may not be as robust as in combination. Suggest changing "ineffective" to "less effective"

6) Some minor errors and/or grammatical suggestions:

Line 34: t(9;22) has been referred as "genes". This is translocation between two "chromosomes", not "genes"

Line 48: there are two "in" 

Line 88: may add a "have" to "that acquired": that have acquired, or change it to "acquire"

Lines 193 and 341: Change word "similarly" to "similar"

Line 258: change % to "percentage", may add % in parenthesis

Line 260: change 0,1 to 0.1

Line 273: Is it supposed to be 0.1% or 0.01%? Also, change "comma" to dot. 

Line 326: change word "commercial" to "industry-sponsored"

Line 353: add a "have" to "TKIs significantly": TKIs have significantly

Line 372: consider changing the word "rapid", for example to "earlier": for earlier TKI switch

- In several places in the manuscript there is an "s" after TKI where the sentence means one single TKI and not plural. Please correct. 

Author Response

This is a well-written well-referenced review of some molecular features of CML and Ph+ ALL along with TKI-directed therapies for these diseases. I congratulate authors for preparing such an organized comprehensive review.

I have a few comments:

  • Suggest considering moving section 4 (starting at line 413) to an earlier part of the manuscript. This section may fit better prior to treatment strategies

Authors: We agree that this change improves the logical flow of the article, therefore we have swapped paragraphs 3 and 4, according to the Reviewer’s suggestion. As insertion of an additional figure at the beginning of the article and reordering the paragraphs caused changes in the line numbering, we updated the line numbers accordingly in the following comments.

  • Line 309-310: Mostly, addition of TKIs to chemotherapy regimens improves long-term outcome (eg, relapse-free survival and OS) in Ph+ ALL and not necessarily/significantly the rate of first complete remission. Majority of patients with Ph+ ALL achieve a complete morphologic remission without TKIs, but they have high relapse rate.

Authors: We agree with the Reviewer that this statement might have been unclear. We wanted to underline that the addition of TKIs to therapy allowed for more patients to undergo allo-HSCT, which improved relapse-free survival and allowed for longer relapse-free survival in patients not fit for allo-HSCT. We changed the whole sentence to “Introduction of TKIs to the therapy improved the long-term outcomes by allowing more patients to undergo allo-HSCT and by achieving durable relapse-free survival without undergoing allo-HSCT in less fit patients.” (Now L399-401).

  • Lines 334, 335, 336: Addition of rituximab to chemotherapy has been mainly studied in Ph- B-cell ALL (Ph+ ALL excluded from studies). Although theoretically rituximab therapy appears effective regardless of Ph status, studies have excluded these patients.

Authors: We propose to remove the fragment concerning rituximab because this issue requires complex discussion as mentioned above (removed from lines 425).

  • Lines 350, 351: Total body irradiation is typically done for allogeneic stem cell transplantation and not for auto-HSCT

Authors: This information was based on 2 publications: Blood. 2003; 104:268a and Blood 2008, 111, (4), 1827-33. We think that to avoid the confusion, the role of auto-HSCT and auto-HSCT with TBI should be precisely discussed, which is beyond the scope of this manuscript. Hence, in the revised version of the manuscript we decided to remove this sentence.

  • Lines 596-597: The statement "these targeted drugs used in monotherapy are ineffective in the management of Ph+ B-ALL". TKI monotherapy can be effective in this disease, but the efficacy may not be as robust as in combination. Suggest changing "ineffective" to "less effective"

Authors: We changed “ineffective” to “less effective” as suggested (now L606-607).

6) Some minor errors and/or grammatical suggestions:

Line 34: t(9;22) has been referred as "genes". This is translocation between two "chromosomes", not "genes"

Authors: We agree, “genes” have been replaced by “chromosomes”(now L35).

Line 48: there are two "in"

Authors: Corrected (now L55).

Line 88: may add a "have" to "that acquired": that have acquired, or change it to "acquire"

Authors: Corrected (now L94).

Lines 193 and 341: Change word "similarly" to "similar"

Authors: Corrected (now L200 and L429).

Line 258: change % to "percentage", may add % in parenthesis

Authors: Corrected (now L347).

Line 260: change 0,1 to 0.1

Authors: Corrected (now L349).

Line 273: Is it supposed to be 0.1% or 0.01%? Also, change "comma" to dot.

Authors: It is supposed to be 0.01% as this is the definition of deep molecular response. Comma was replaced with a dot (now L361).

Line 326: change word "commercial" to "industry-sponsored"

Authors: Corrected (now L417).

Line 353: add a "have" to "TKIs significantly": TKIs have significantly

Authors: Corrected (now L440).

Line 372: consider changing the word "rapid", for example to "earlier": for earlier TKI switch

Authors: We agree, corrected to “earlier” (now L459).

- In several places in the manuscript there is an "s" after TKI where the sentence means one single TKI and not plural. Please correct.

Authors: We have found this error in one place, in a section which has been removed.

Reviewer 3 Report

This MS by Komorowski et al is a thoughtful compilation of the recent literature on important subtypes of leukemia, and thus likely to be useful to researchers formulating plans for both laboratory and clinical studies of leukemia. Many of the publications are from the last five years and are summarized in four Tables and one  two-part figure.

 Although the great predominance of the text is well written and clear,  some  clarifications  would  enhance its  readability and result in a clearer picture of the current state of the science of these diseases.

  1. The key “actor” in this review is the Philadelphia chromosome, referenced briefly on line 35 as a 1962 publication in “Blut”. A   few sentences should  be added  explaining the discovery  of this small abnormal chromosome by Peter Nowell, was first published in a brief paper/abstract in “Science {142:1497,1960]”, with the help of  David Hungerford. Both being in Philadelphia ( U of P,  and the  Institute for Cancer Research, respectively) Peter modestly coined the phrase “Philadelphia chromosome  “. 
  2. On line 79 , what is meant by “1.95 out of 13.9 new cases? Hard to compare with the European number.
  3. L 220. Can the first sentence be bolded to emphasize the content?
  4. L 495-496 “ CML in LBP” . Meaning ?
  5. L 597 .  Give examples of the “ intensive chemotherapy”. Targeting TKIs is also chemotherapy, and may be intensive.
  6. L 616. “TXNIP ” is mentioned as a “glucose transport inhibitor(s)”. This may be misleading. The thioredoxin-interacting protein is principally known as a Redox regulator, and as such also regulates apoptosis. However, since it was first identified as a Vitamin D Receptor Interacting Protein, it is likely that it can also interact with other proteins, including those that regulate glucose transport.  This should  be discussed, before moving on to Fig 1B.
  7. L 622. Where is Fig 1A mentioned? The discussion of Fig1A should be preceded by making clear that this is what is being discussed, and is quite comprehensive. However, the discussion of Fig A2 and A3 is hard to find. This is a major omission.

Author Response

This MS by Komorowski et al is a thoughtful compilation of the recent literature on important subtypes of leukemia, and thus likely to be useful to researchers formulating plans for both laboratory and clinical studies of leukemia. Many of the publications are from the last five years and are summarized in four Tables and one  two-part figure.

 Although the great predominance of the text is well written and clear,  some  clarifications  would  enhance its  readability and result in a clearer picture of the current state of the science of these diseases.

  1. The key “actor” in this review is the Philadelphia chromosome, referenced briefly on line 35 as a 1962 publication in “Blut”. A   few sentences should  be added  explaining the discovery  of this small abnormal chromosome by Peter Nowell, was first published in a brief paper/abstract in “Science {142:1497,1960]”, with the help of  David Hungerford. Both being in Philadelphia ( U of P,  and the  Institute for Cancer Research, respectively) Peter modestly coined the phrase “Philadelphia chromosome  “. 

Authors: We thank the reviewer for this suggestion. We included the additional information and the corresponding reference in the revised version of the manuscript (L37-38).

  1. On line 79 , what is meant by “1.95 out of 13.9 new cases? Hard to compare with the European number.

Authors: We agree that the sentence in the previous version of the manuscript was unclear and not comparable with the European data. In the revised version of the manuscript we have changed the US statistics to be more consistent with European one (now L85).

  1. L 220. Can the first sentence be bolded to emphasize the content?

Authors: Figure and table captions are now bolded, according to the suggestion (L47, 228, 329, 390, 409, 652).

  1. L 495-496 “ CML in LBP” . Meaning ?

Authors: CML in LBP stands for CML in lymphoid blast phase (now L503). This abbreviation is explained earlier in the text – L119.

  1. L 597. Give examples of the “ intensive chemotherapy”. Targeting TKIs is also chemotherapy, and may be intensive.

Authors: To be more precise, we have replaced “intensive” with “other” (now L 607). We decided not to specifically name the drugs in the “Conclusions” section, as they are already discussed earlier, in the section “4.2. TKIs in the treatment of Ph+ ALL.”

  1. L 616. “TXNIP ” is mentioned as a “glucose transport inhibitor(s)”. This may be misleading. The thioredoxin-interacting protein is principally known as a Redox regulator, and as such also regulates apoptosis. However, since it was first identified as a Vitamin D Receptor Interacting Protein, it is likely that it can also interact with other proteins, including those that regulate glucose transport.  This should  be discussed, before moving on to Fig 1B.

Authors: We agree that referring to TXNIP as a glucose transport inhibitor might be misleading. As the detailed discussion of the redox regulation by the thioredoxin family of proteins is beyond the scope of the article, to avoid confusion we added “direct and indirect” part to “glucose transport inhibitors”  (L548, L627-628).

  1. L 622. Where is Fig 1A mentioned? The discussion of Fig1A should be preceded by making clear that this is what is being discussed, and is quite comprehensive. However, the discussion of Fig A2 and A3 is hard to find. This is a major omission.

Authors: To improve clarity of the Figure description (Fig. 2A in the revised version of the manuscript), we refer to the specific parts of the figure in the text (L622-623, L626, L629-630). We also added the sentence: “The figure description is in the Conclusions section” in the figure legend, L652-653.

Reviewer 4 Report

1. I think the abstract perfectly covers the main highlights of this literature review.

2. Starting from section-4 is where I feel genuinely interested in reading.

3. Based on your abstract and conclusions, I believe the development of therapies on "Ph+ B-ALL" is what the authors emphasize, which makes the title unrelated, unspecific, not very consistent with the full text.

3. I would suggest concentrating the section-2 into the introduction part. The "p190 and p210 BCR-ABL variants" part could begin earlier so that the shortcomings of current TKI therapies on "Ph+ B-ALL" can be followed more directly; then the story will be lead to the challenge of current TKI therapies on "Ph+ B-ALL", the development of new treatments on "Ph+ B-ALL".

It will be much better if it can be re-organized. Thank you!

Author Response

  1. I think the abstract perfectly covers the main highlights of this literature review.
  2. Starting from section-4 is where I feel genuinely interested in reading.
  3. Based on your abstract and conclusions, I believe the development of therapies on "Ph+ B-ALL" is what the authors emphasize, which makes the title unrelated, unspecific, not very consistent with the full text.

Authors: We agree with the Reviewer that the title did not adequately reflect the content of the article. We proposed a new title:

“Philadelphia chromosome-positive leukemia in the lymphoid lineage – similarities and differences with the myeloid lineage and specific vulnerabilities”

  1. I would suggest concentrating the section-2 into the introduction part. The "p190 and p210 BCR-ABL variants" part could begin earlier so that the shortcomings of current TKI therapies on "Ph+ B-ALL" can be followed more directly; then the story will be lead to the challenge of current TKI therapies on "Ph+ B-ALL", the development of new treatments on "Ph+ B-ALL". It will be much better if it can be re-organized. Thank you!

Authors: We agree that the article will benefit from some re-organization. Following the Reviewers’ suggestion, we moved the paragraph on p190 and p210 BCR-ABL variants before the paragraph on TKIs and treatment shortcomings. However, we decided not to incorporate paragraph 2 discussing mutational landscapes of CML and B-ALL into the introduction. Paragraph 2 is very comprehensive and we believe that this level of detail is important for a better understanding of the following paragraphs, especially paragraph 5 which summarizes vulnerabilities of Ph+ B-ALL. We believe that narrowing it down may reduce the clarity of the entire article.

Round 2

Reviewer 4 Report

I agree with the revisions. Thanks for the timely response.